# Autologous Collagen-Induced Chondrogenesis: From Bench to Clinical Development

**DOI:** 10.3390/medicina59030530

**Published:** 2023-03-08

**Authors:** You Seung Chun, Seon Ae Kim, Yun Hwan Kim, Joong Hoon Lee, Asode Ananthram Shetty, Seok Jung Kim

**Affiliations:** 1Department of Orthopedic Surgery, College of Medicine, The Catholic University of Korea, Seoul 06591, Republic of Korea; 2Institute of Medical Sciences, Faculty of Health and Wellbeing, Canterbury Christ Church University, Chatham Maritime, Kent ME4 4UF, UK

**Keywords:** atelocollagen, microfracture, cartilage repair, knee, ACIC

## Abstract

Microfracture is a common technique that uses bone marrow components to stimulate cartilage regeneration. However, the clinical results of microfracture range from poor to good. To enhance cartilage healing, several reinforcing techniques have been developed, including porcine-derived collagen scaffold, hyaluronic acid, and chitosan. Autologous collagen-induced chondrogenesis (ACIC) is a single-step surgical technique for cartilage regeneration that combines gel-type atelocollagen scaffolding with microfracture. Even though ACIC is a relatively new technique, literature show excellent clinical results. In addition, all procedures of ACIC are performed arthroscopically, which is increasing in preference among surgeons and patients. The ACIC technique also is called the Shetty–Kim technique because it was developed from the works of A.A. Shetty and S.J. Kim. This is an up-to-date review of the history of ACIC.

## 1. Introduction

The articular cartilage is a thin layer of viscoelastic connective tissue that complicates repair of the cartilage. Studies have emphasized the importance of repairing articular cartilage to delay arthroplasty. Multiple drilling techniques have been used to treat articular cartilage defects. In 1959, Pridie [1] said, “If these sclerotic areas were drilled and the holes were not too far apart, smooth fibro-cartilage would spread over the surface.” He introduced the drilling of the subchondral bone to heal cartilage defects.

Steadman et al. [2] modified this method to avoid thermal necrosis from drilling, using bone marrow from the subchondral bone to form a clot on the defect holes (Figure 1). The clot contained mesenchymal stem cells (MSCs) and abundant growth factors, inducing repair by fibrous and hyaline-like cartilage [3] (Figure 2).

Multiple drilling techniques produce predominantly fibrocartilage [4]; therefore, several methods have been introduced to produce hyaline-like cartilage. Autologous chondrocyte implantation is considered an ideal procedure to induce hyaline-like cartilage. However, it requires a two-stage procedure, damages the donor site cartilage, and has a high cost.

The limitation of the marrow stimulation technique is that bone marrow stem cells and growth factors are released into the joint rather than remaining at an articular surface. Collagen gel or scaffold has been used to overcome this limitation by providing mechanical stability to form clots [5]. Cell-free type I collagen gels or scaffolds combined with marrow stimulation, such as multiple drilling, have shown good outcomes regarding the induction of hyaline-like cartilage [6,7,8]. Autologous matrix-induced chondrogenesis (AMIC) uses a porcine collagen matrix to provide a biological scaffold.

Despite its popularity since 2000, the AMIC technique requires an open surgical incision for the preparation of the defect and the application of the collagen membrane. To overcome this weakness, autologous collagen-induced chondrogenesis (ACIC) was developed, for which the procedures are performed arthroscopically. For ACIC, atelocollagen gel is used as a scaffold instead of the membrane used in AMIC. Atelocollagen is a highly purified type I collagen obtained following the treatment of skin dermis with pepsin and telopeptide removal, which reduces immunogenicity.

This article reviews the developmental history with a basic scientific rationale and the various techniques and results of ACIC for the repairing of knee cartilage.

## 2. Basic Science and Methods for Chondrogenesis

### 2.1. Basic Science of Cartilage Injury

Articular cartilage is a thin, viscoelastic layer of connective tissue 2–3 mm thick [9] (Figure 3). It is of mesodermal origin and is characterized by a cellular component immersed within an extracellular matrix composed of polysaccharides, fibrous protein, and interstitial fluid [10]. The cartilage has no direct supply of blood, nerve signals, or nutrition and relies on diffusion through the surrounding tissues.

Cartilage can be damaged easily by acute trauma or repetitive microtrauma and is exposed to mechanical stress during active daily living. Such stress can be increased to 10 or 20 times the body weight during sports activities [9]. Acute cartilage injury initiates a repair process that starts with the formation of a blood clot containing bone marrow cells that form fibrocartilaginous tissue. However, repeated microtraumas damage chondrocytes, decrease production of proteoglycan, and damage collagen meshwork. As this meshwork limits water penetration, damage leads to swelling and stiffness of the tissue. Responses to cartilage injury involve both anabolic and catabolic reactions. Aggrecan-degrading enzyme disintegrin, metalloproteinase with thrombospondin motif 5 (ADAMTS-5), and collagenase matrix metalloproteinase 13 (MMP-13), which degrades type II collagen, all contribute to the breakdown of cartilage. However, the induction of chondroprotective genes leads to anabolic effects on the cartilage [11] (Figure 4).

### 2.2. Microfracture and Enhanced Microfracture

Microfracture is a minimally invasive technique that uses an arthroscope to drill small, equidistant holes in subchondral bone, at least 3–4 mm apart and 4 mm in depth, with 3–4 holes per a 1 cm area [12]. This procedure induces the migration of MSCs from the bone marrow to the cartilage defect to allow the formation of fibrocartilage [13]. Kruez et al. reported that microfracture showed good short-term results in small cartilage defects but poorer results at 18 months after surgery, as reflected by Cincinnati Knee Rating System and International Cartilage Repair Society (ICRS) scores. This effect is clearer in large defects and in defects of the patellofemoral joint [14], where subchondral osteophytes develop in 20–50% of cases [15].

AMIC adds a collagen scaffold to multiple drilled holes in a cartilage defect. The scaffold of the collagen binds MSCs and growth factors to the cartilage defect, enhancing regeneration [16,17]. MSCs from microfracture have the same phenotypic plasticity as chondrogenic cells in the cartilage basal zone. With AMIC, MSCs are distributed on the membrane, which acts as the roof of a “biological chamber” [18]. A systematic review conducted in 2022 by Migliorini et al. reported that AMIC showed better clinical scores and a lower rate of revision, compared to microfracture [19].

AMIC requires an open procedure to attach the collagen matrix to the cartilage defect (Figure 5). Even a small defect in articular cartilage needs a large incision for the AMIC procedure, which can delay patient recovery.

To overcome this, ACIC was introduced as an alternative in 2009. Together, Shetty and Kim developed the ACIC technique to be performed with arthroscopy, leaving wounds only for the arthroscopic portal incisions [20]. In this procedure, atelocollagen gel is mixed with fibrin and thrombin and used as a scaffold. The mixture is applied on the cartilage defect after microdrilling the defect under CO_2_ gas insufflation to maintain the collagen clot in the chondral defect area.

### 2.3. Autologous Collagen-Induced Chondrogenesis (ACIC)

#### 2.3.1. Basic Science of ACIC

Atelocollagen is a porcine type I collagen that has been treated to detach the telomeres (Figure 6). This process suppresses immunologic reactions, increasing the effects of the surrounding environment on the collagen.

Atelocollagen induces MSCs to differentiate into chondrocytes (Figure 7). The process is examined by the expression of genes such as Sox9, type II collagen, and aggrecan. Jeong et al. conducted an animal study with rabbits involving a circular, articular cartilage defect 4 mm in diameter in the trochlear region. The 10 rabbits in the control group were not treated, and the 10 rabbits in the experimental group underwent injections of atelocollagen mixed with fibrin. After 12 weeks, the cartilage was examined. The experimental group had regenerated smooth, hyaline-like cartilage, whereas the control group showed incomplete and irregular cartilage [21] (Figure 8).

Many clinical trials have used atelocollagen as an enhancing material for microfractures in a cartilage defect [20,21,22]. A recent study showed that atelocollagen promotes the chondrogenic differentiation of human adipose-derived MSCs. Chondrogenic genes and proteins were evaluated by RT-qPCR and ELISA, showing differentiation on day 21 [23]. MRI follow-up was conducted at 1 year and showed good cartilage defect filling [20].

#### 2.3.2. Surgical Technique

ACIC is performed in the same manner as routine knee arthroscopy (Figure 9). The articular cartilage is evaluated and mapped according to ICRS guidelines. The lesion is debrided until the margins show stable vertical cartilage. The surface is lightly abraded with a curette or burr to debride sclerotic subchondral bone. Microdrilling is performed, and the subchondral bone is drilled with a 3.5 mm diameter bit up to a depth of 6–10 mm, with a 3 mm interval. After that, the arthroscopic water of the knee joint is drained, and the joint is insufflated with CO_2_ (Figure 9). The surface of the subchondral bone is dried with cotton to promote adhesion by the collagen mixture. Atelocollagen gel mixed with fibrinogen and thrombin (ratio—fibrin 1: thrombin 0.2 and atelocollagen 0.8) is prepared and applied under arthroscopy. The applied mixture is assessed for stability in the cartilage defect by observing the range of motion of the knee after 2 min [20,24] (Figure 10).

#### 2.3.3. Post-Operative Protocol

Patients are recommended to undergo continuous passive motion (CPM) rehabilitation for 4 h/day post-operatively for 4–6 weeks. Patients with a femoral condyle lesion are allowed partial weight-bearing at 6 weeks, and progression to full weight-bearing is encouraged at 3 months. Cartilage being repaired in patellar and trochlear lesions is protected with a knee brace locked at 0–20% movement, which is gradually increased to 90% over 6 weeks. Full weight-bearing is encouraged with protected knee motion right after the surgery [24].

#### 2.3.4. Clinical Results

Clinical studies of ACIC were collected through PubMed, the Cochrane library, and ScienceDirect, with keywords “autologous collagen-induced chondrogenesis” and “porcine, collagen, chondrogenesis”; there have been several studies using ACIC on the knee or the talus. Excluding studies about the talus, five studies were analyzed. Two studies were randomized, controlled trials (RCTs), one study was a matched, comparative study, and two were noncomparative studies. (Table 1)

Kim M.S. et al. compared porcine-derived collagen-augmented chondrogenesis to microfracture in 2020 in a multicenter, randomized control study. One hundred patients were randomly assigned to a microfracture or an investigational group. Clinical and MRI outcomes were assessed at 12 and 24 months post-operatively. Magnetic resonance observation of cartilage repair tissue (MOCART) assessment was used to analyze cartilage tissue repair. MOCART score, VAS score, and KOOS pain score were significantly improved in the test group. In addition, the investigational group showed better filling of the cartilage defect in the knee joint [25].

Silva et al. compared ACIC to microfracture in 2020 based on clinical scores at 6 months and 24 months. Eleven patients who underwent ACIC were compared with 11 age- and sex-matched patients who underwent a microfracture-only procedure. The ACIC group showed a significantly better SF36 mental function, International Knee Documentation Committee (IKDC) score, and VAS score at 24 months [26].

Kim S.J. et al. evaluated 30 patients with ICRS grade III/IVa symptomatic knees who were treated with ACIC. Patients were followed for 6 years, and the Lysholm score, Knee Injury and Osteoarthritis Outcome Score (KOOS), and IKDC score were significantly improved. Radiological evaluation was performed using MRI at 6 months, 1 year, and 3 years using the MOCART scoring system. The mean MOCART score was 78, similar to those of other successful cartilage repair techniques [24].

Shetty et al. evaluated 10 patients with symptomatic chondral defects who were treated with ACIC. Morphological and biochemical MRIs were performed at the 1-year follow-up, and the Lysholm score was assessed at the 2-year follow-up. MRIs showed good cartilage defect filling and suggested hyaline-like repair tissue, and the Lysholm score was significantly improved [20].

## 3. Discussion

Mithoefer et al. conducted a systematic review about microfracture in 2009. Microfracture was a good first-line treatment for cartilage defect but did have a few disadvantages. The short-term outcome was good, but the long-term outcome was inconclusive, due to insufficient data. Shortcomings of microfracture included limited regeneration of hyaline-like cartilage, variable volume of cartilage repair, and functional deterioration [15]. As a consequence, many cartilage regeneration techniques were designed to overcome these limitations of microfracture. ACIC not only improved short-term clinical outcomes, it also significantly improved long-term clinical outcomes at 6 years [24].

ACIC is one of the methods for enhancing cartilage repair; it has no donor site comorbidity and is performed using arthroscopy in a single stage. An animal study and an in vitro study showed that atelocollagen promotes hyaline-like cartilage regeneration [21,23]. These encouraging outcomes and the concept of a single-stage cartilage resurfacing technique are attractive for many surgeons.

Though ACIC is performed mostly under arthroscopy, one study involved a mini open procedure. A multicenter study by Kim M.S. et al. involved an open procedure applying atelocollagen at the chondral defect, whereas S Kim S.J. used CO_2_ gas to inflate the joint space while applying atelocollagen. CO_2_ gas allows the gravity-independent application of the gel mixture without opening the joint. Thus, all procedures can be conducted under arthroscopy, minimizing soft tissue damage. The method of microfracture may influence the outcomes of cartilage regeneration. Kim S.J. used a 3.5 mm diameter drill up to a depth of 10 mm, with a 3 mm interval. On the other hand, the multicenter study of Kim M.S. did not specify the technique of microfracture. A fibrin and thrombin mixture is used to stabilize the collagen scaffold; however, the mixture ratio varies among studies. Since the collagen scaffold plays a major role in the good outcome of the microfracture technique, the durability of the collagen scaffold may influence the outcomes of cartilage regeneration.

Clinical trials about ACIC are limited. Silva et al. compared ACIC with microfracture in an RCT, another of which was performed by Kim M.S. et al. In addition, Kim S.J. et al. and Shetty et al. found better clinical results and superior MRI findings to microfracture.

This review showed the need to standardize reporting of the AMIC technique to enable future comparisons of efficacy and determine the effects of various technical variations [5]. Some of the technical factors that should be reported are as follows:(1)Arthroscopic or open surgery(2)Method of subchondral drilling or microfracture(3)Type of gel used(4)Fixation of the matrix or scaffold(5)Postoperative rehabilitation

In addition, we suggest that the measures for outcomes be standardized. The most common outcome instruments are KOOS, Lysholm, and IKDC scoring. We suggest that these three instruments be used in studies with follow-ups of at least 1–2 years for adequate comparison. MRI protocols for cartilage assessment, such as the modified MOCART score suggested by Marlovits et al., [27], should be routine to allow the comparison of follow-up radiographic studies.

An advantage of ACIC is its cost-effectiveness, compared to ACI. In the UK, the cost of ACI is three times that of ACIC [28]. In addition, ACIC has no donor site morbidity and is performed in one stage.

There are some limitations of ACIC. The few prospective studies of ACIC each had a small sample size. While the results were positive and showed the usefulness of ACIC, comparison studies to other enhanced microfracture techniques are needed.

## 4. Conclusions

Our review of the ACIC technique suggests it is a promising cartilage repair technique. The outcome scores and MRI results are promising, but there are few comparative studies with other cell-based cartilage methods. Ideal conditions for chondrogenesis should be studied.

## Figures and Tables

**Figure 1 medicina-59-00530-f001:**
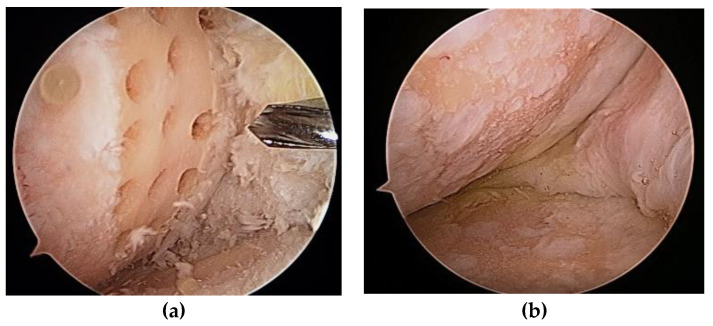
(**a**) Multiple holes drilled into a cartilage defect; (**b**) second-look arthroscopy 2 years later.

**Figure 2 medicina-59-00530-f002:**
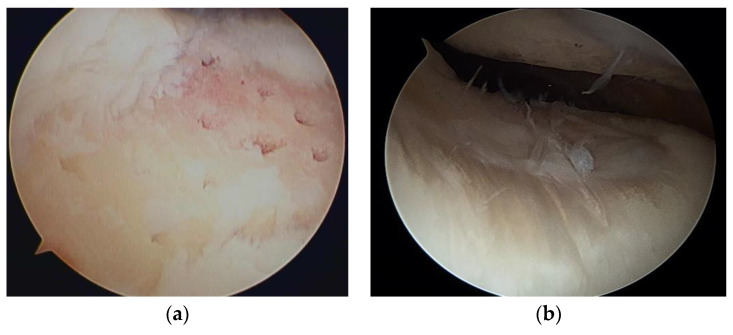
(**a**) Microfracture of a cartilage defect of the trochlea; (**b**) second-look arthroscopy 2 years later.

**Figure 3 medicina-59-00530-f003:**
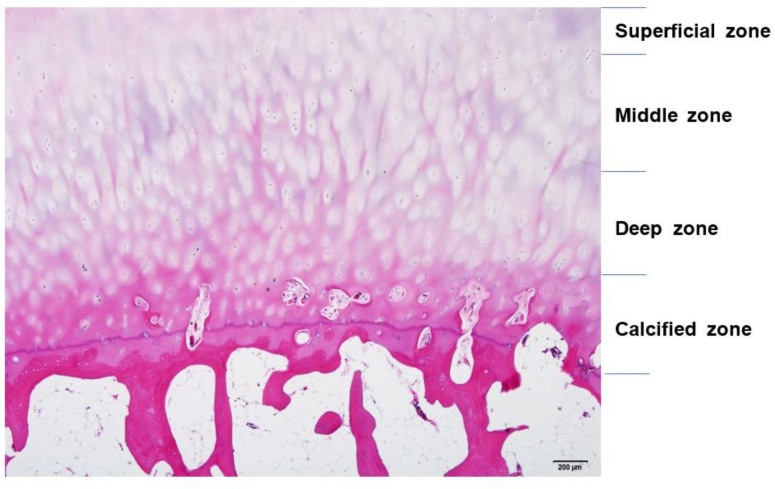
The structure of articular cartilage, comprised of four zones: superficial, middle, deep, and calcified. Among zones, there are differences in collagen fibers, arrangement of chondrocytes, and distribution of proteoglycans and glycosaminoglycans (GAGs). Hematoxylin and eosin staining. Scale bar, 200 μm.

**Figure 4 medicina-59-00530-f004:**
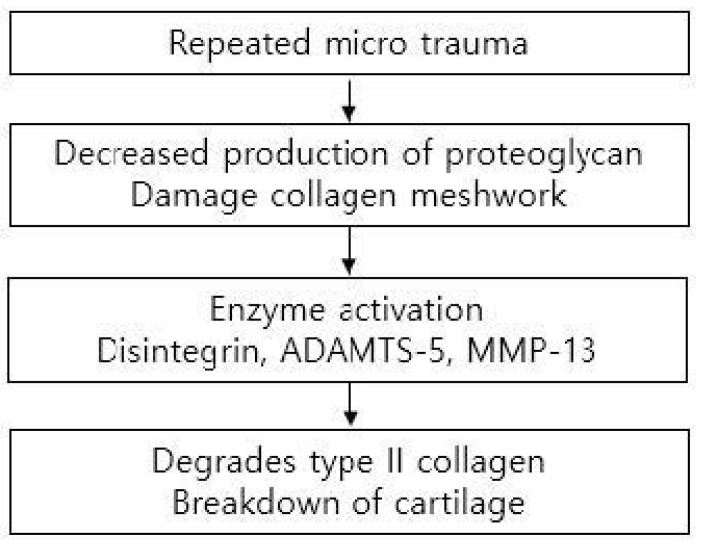
Mechanism leading to the breakdown of cartilage.

**Figure 5 medicina-59-00530-f005:**
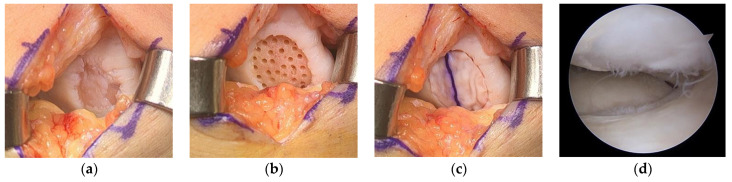
(**a**) Cartilage defect, (**b**) after defect preparation, (**c**) membrane-covered defect, and (**d**) second-look arthroscopy 2 years later. Image courtesy of Prof. Sung-Hwan Kim.

**Figure 6 medicina-59-00530-f006:**
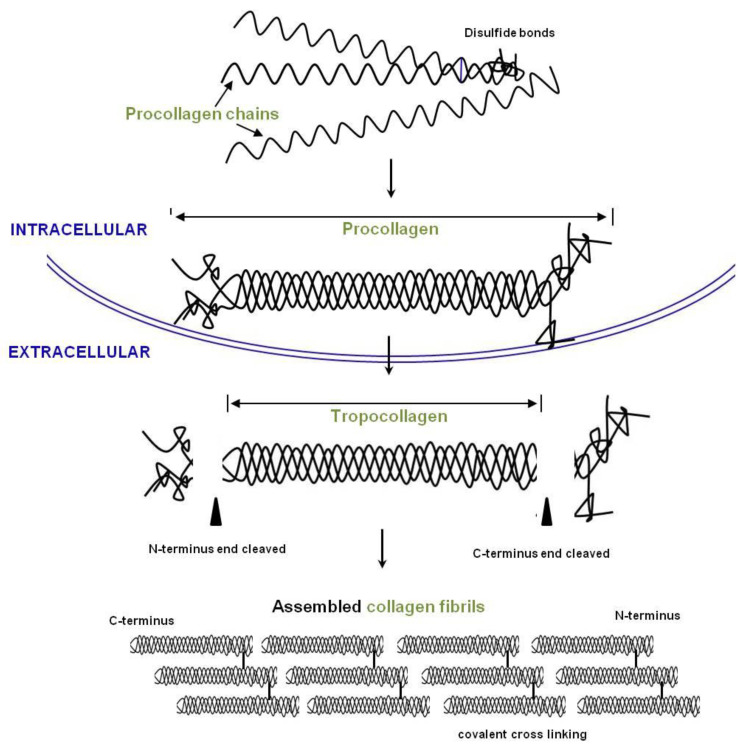
Overview of collagen production. Procollagen chains are synthesized and form a helix in the endoplasmic reticulum (ER). Propeptides are removed by proteinases, and the fibrils are assembled into collagen fibers in the extracellular space (EC).

**Figure 7 medicina-59-00530-f007:**
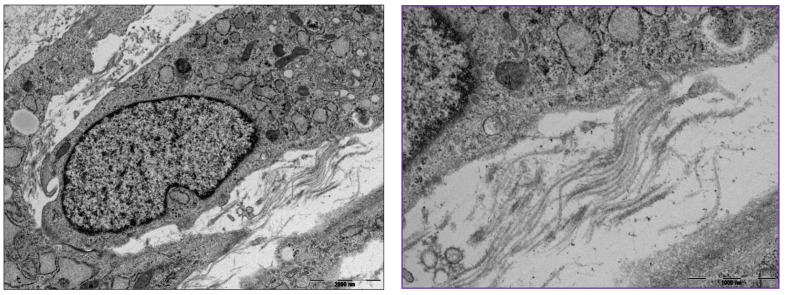
Transmission electron microscopy of differentiated mesenchymal stem cells (MSCs) into chondrocytes and extracellular collagen fibrils (box). (**Left**) Scale bar, 2000 nm; (**Right**) scale bar, 1000 nm.

**Figure 8 medicina-59-00530-f008:**
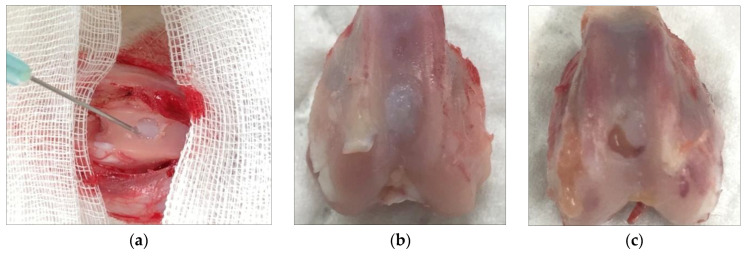
(**a**) Injection of atelocollagen mixed with fibrin into the trochlear defect of a rabbit knee. (**b**) Atelocollagen injection group after 12 weeks. (**c**) Control group without injection after 12 weeks.

**Figure 9 medicina-59-00530-f009:**
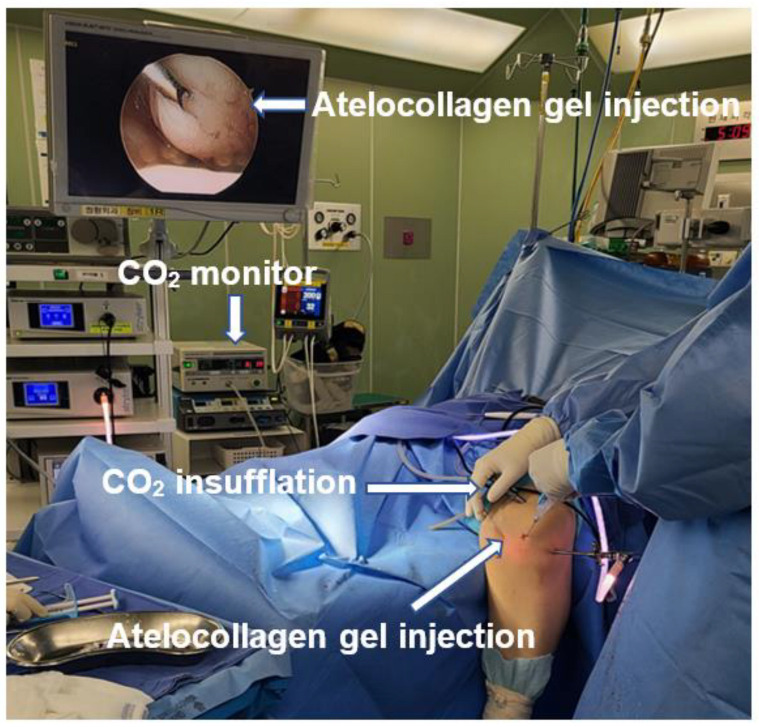
Arthroscopic settings for ACIC.

**Figure 10 medicina-59-00530-f010:**
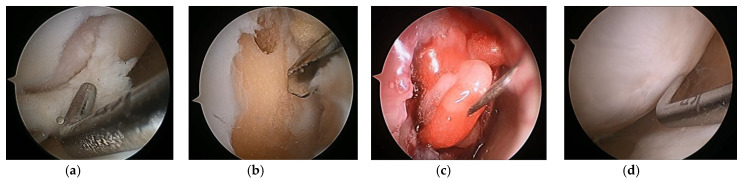
(**a**) Cartilage defect, (**b**) defect preparation and multiple drilling, (**c**) collagen gel injection into the defect under CO_2_ gas insufflation, and (**d**) second-look arthroscopy at 2 years later.

**Table 1 medicina-59-00530-t001:** Summary of clinical studies on ACIC for knee chondral defects.

Authors	No. of Patients	Study Design	Cohort Group	Clinical Scores	MRI Evaluation
Kim, M.S. [25]	100	Multicenter RCT	Microfracture	KOOS pain, VAS; significant difference	MOCART
Silva [26]	11	Comparative study	Microfracture	SF-36, IKDC; significant difference	none
Kim, S.J. [24]	30	Longitudinal study		Lysholm, KOOS, IKDC	MOCART
Shetty [20]	10	Longitudinal study		Lysholm	MOCART

KOOS: Knee Injury and Osteoarthritis Outcome Score, MOCART: Magnetic Resonance Observation of Cartilage Repair Tissue, IKDC: International Knee Documentation Committee.

## Data Availability

Previous studies included in this study can be found on PubMed.

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
