# Peer review of "Autologous Collagen-Induced Chondrogenesis: From Bench to Clinical Development"

_medicina, 2023, doi:10.3390/medicina59030530_

Round 1
Reviewer 1 Report
Microfracture is a common technique that uses bone marrow components to stimulate cartilage regeneration. However, the clinical results of microfracture range from poor to good. To enhance cartilage healing, several reinforcing techniques have been developed, including the autologous matrix-induced chondrogenesis (AMIC), autologous collagen-induced chondrogenesis (ACIC), and so on.
This article focuses on the ACIC technique with basic scientific rationale, operative protocol and clinical results. It also reviews the developmental history of various techniques and their advantages and limitations for repair of knee cartilage.
In conclusion, this review gives readers a complete and deep knowledge on ACIC technique and is suggested to publish after minor revision.
1. some pictures are not very clear, change the numbering format to another style, such as 1) , in line 272-275.
2. Add number 2 and the subtitle before line 70.
Author Response
Thank you for your comment.
We inserted images with higher dpi for figure 6 and 7. For other images, current image has the finest resolution.
We added a subtitle and numbering for methods for chondrogenesis.
“2. Basic science and Methods for Chondrogenesis”
Also, we changed numbering style for five technical factors.
“1) Arthroscopic or open surgery
2) Method of subchondral drilling or microfracture
3) Type of gel used
4) Fixation of the matrix or scaffold
5) Postoperative rehabilitation “
Reviewer 2 Report
I have read the review paper with title: Autologous collagen-induced chondrogenesis: From bench to clinical development.
The paper’s subject is very interesting because it deals with important problem in daily clinical practice.
The aim of the author’s work was to present curent state in the field of chondrogenesis and analysis of results in few relevant studies about autologous collagen-induced chondrogenesis. Their paper is well organised with clear presented methodology and results, very ilustrative figures.
In my opinion, the paper deserves to be published.
Authors should correct just one technical issues:
- I recommend them to mark in text number of the figures which are in relation with some text.
Best regards!
Author Response
Thank you for your comment.
We marked text number of the figures which are in relation with text for figure 1 to 9.
Reviewer 3 Report
I sustain that this article can be published in "Medicina" Journal. Nice figures, interesting information. Nice to see that the ACIC technique it is a promising cartilage repair technique that have a promising MRI outcome score.
Author Response
Thank you for your comment.